# A Structured Prediction Approach for Generalization in Cooperative Multi-Agent Reinforcement Learning

**Nicolas Carion**
Facebook, Paris
Lamsade, Univ. Paris Dauphine
alcinos@fb.com

**Gabriel Synnaeve**
Facebook, NYC
gab@fb.com

**Alessandro Lazaric**
Facebook, Paris
lazaric@fb.com

**Nicolas Usunier**
Facebook, Paris
usunier@fb.com

## Abstract

Effective coordination is crucial to solve multi-agent collaborative (MAC) problems. While centralized reinforcement learning methods can optimally solve small MAC instances, they do not scale to large problems and they fail to generalize to scenarios different from those seen during training. In this paper, we consider MAC problems with some intrinsic notion of locality (e.g., geographic proximity) such that interactions between agents and tasks are locally limited. By leveraging this property, we introduce a novel structured prediction approach to assign agents to tasks. At each step, the assignment is obtained by solving a centralized optimization problem (the inference procedure) whose objective function is parameterized by a learned scoring model. We propose different combinations of inference procedures and scoring models able to represent coordination patterns of increasing complexity. The resulting assignment policy can be efficiently learned on small problem instances and readily reused in problems with more agents and tasks (i.e., zero-shot generalization). We report experimental results on a toy search and rescue problem and on several target selection scenarios in StarCraft: Brood War[1], in which our model significantly outperforms strong rule-based baselines on instances with 5 times more agents and tasks than those seen during training.

## 1 Introduction

Multi-agent collaboration (MAC) problems often decompose into several intermediate tasks that need to be completed to achieve a global goal. A common measure of size, or difficulty, of MAC problems is the number of agents and tasks: more tasks usually require longer-term planning, the joint action space grows exponentially with the number of agents, and the joint state space is exponential in both the numbers of tasks and agents. While general-purpose reinforcement learning (RL) methods [26] are theoretically able to solve (centralized) MAC problems, their learning (e.g., estimating the optimal action-value function) and computational (e.g., deriving the greedy policy from an action-value function) complexity grows exponentially with the dimension of the problem. A way to address this limitation is to learn in problems with few agents and a small planning horizon and then *generalize* the solution to more complex instances. Unfortunately, standard RL methods are not able to perform any meaningful generalization to scenarios different from those seen during training. In this paper we study problems whose structure can be exploited to learn policies in small instances that can be efficiently generalized across scenarios of different size.

Well-known MAC problems that are solved by a suitable sequence of agent-task assignments include search and rescue, predator-prey problems, fleet coordination, or managing units in video games. In all these problems, the dynamics describing the interaction of the "objects" in the environment (i.e., agents and tasks) is regulated by *constraints* that may greatly simplify the problem. A typical example is *local proximity*, where objects' actions may only affect nearby objects (e.g., in the predator-prey, the prey's movements only depend on nearby agents). Similarly, constraints may be related to *assignment proximity*, as agents may only interact with agents assigned to the same task.

The structure of problems with *constrained interaction* has been exploited to simplify the learning of value functions [e.g., 20] or dynamics of the environment [e.g., 11]. These approaches effectively generalize from easier to more difficult instances: we may train on small environments where the sample complexity is practical and generalize to large problems without ever training on them (*zero-shot generalization*). The main drawback is that when generalizing value functions or the dynamics, the optimal (or greedy) policy still needs to be recomputed at each new instance, which usually requires solving an optimization problem with complexity exponential in the number of objectives (e.g., maximizing the action-value function over the joint action space).

In this paper, we build on the observation that in MAC problems with constrained interaction, optimal policies (or good approximations) can be effectively represented as a combination of coordination patterns that can be expressed as reactive rules, such as creating subgroups of agents to solve a single task, avoiding redundancy, or combinations of both. We decompose agents' policies into a high-level agent-task assignment policy and a low-level policy that prescribes the actual actions agents should take to solve the assigned task. As the most critical aspect of MAC problems is the coordination between agents, we assume low-level policies are provided in advance and we focus on learning effective high-level policies. To leverage the structure of the assignment policy, we propose a structured prediction approach, where agents are assigned to tasks as a result of an optimization problem. In particular, we distinguish between the *coordination inference procedure* (i.e., the optimization problem itself) and *scoring models* (the objective function) that provide a score to agent-task and task-task pairs. In its more complex instance, we define a quadratic inference procedure with linear constraints, where the objective function uses learned pairwise scores between agents and tasks for the linear part of the objective, and between different tasks for the quadratic part. With this structure we address the intrinsic exponential complexity of learning in large MAC problems through *zero-shot generalization*: **1)** the parameters of the scoring model can be learned in small instances, thus keeping the learning complexity low, **2)** the coordination inference procedure can be generalized to an arbitrary number of agents and tasks, as its computational complexity is polynomial in the number of agents and tasks. We study the effectiveness of this approach on a search and rescue problem and different battle scenarios in "StarCraft: Brood War". We show that the linear part of the optimization problem (i.e., using agent-task scores) represents simple coordination patterns such as assigning agents to their closest tasks, while the quadratic part (i.e., using task-task scores) may capture longer-term coordination such as spreading the different agents to tasks that are far away to each other or, on the contrary, create groups of agents that focus on a single task.

## 2 Related Work

Multi-agent reinforcement learning has been extensively studied, mostly in problems of decentralized control and limited communication (see Busoniu et al. [2] for a survey). By contrast, this paper focuses on *centralized control under full state observation*.

Our work is closely related to generalization in relational Markov Decision Processes [11] and decomposition approaches in loosely and weakly coupled MDPs [24, 17, 10, 27, 20]. The work on relational MDPs and the related object-oriented MDPs and first-order MDPs [11, 6, 22] focus on learning and planning in environments where the state/action space is compactly described in terms of objects (e.g., agents) that interact with each other, without prior knowledge of the actual number of objects involved. Most of the work in this direction is devoted to either efficiently estimating the environment dynamics, or approximate the planning in new problem instances. Whereas the type of environments and problems we aim at are similar, we focus here on model-free learning of policies that generalize to new (and larger) problem instances *without* replanning.

Loosely or weakly coupled MDPs are another form of structured MDPs, which decompose into smaller MDPs with nearly independent dynamics. These works mostly follow a decomposition approach in which global action-value functions are broken down into independent parts that are either learned individually, or serve as guide for an effective parameterization for function approximation.

The policy parameterization we develop follows the task decomposition approach of Proper and Tadepalli [20], but the policy structures we propose are different. Proper and Tadepalli [20] develop policies based on pairwise interaction terms between tasks and agents similar to our quadratic model, but the pairwise terms are based on interactions dictated by the dynamics of the environment (e.g., agent actions that directly impact the effect of other actions) aiming at a better estimation of the value function of low-level actions of the agents once an assignment is fixed, whereas our quadratic term aims at assessing the long-term value of an assignment.

Many deep reinforcement learning algorithms have been recently proposed to solve MAC problems with a variable number of agents, using different variations of communication and attention over graphs [25, 7, 32, 31, 13, 12, 16, 23]. However, most of these algorithms focus on fixed-size action spaces, and little evidence has been given that these approaches generalize to larger problem instances [28, 8, 32]. Rashid et al. [21] and Lin et al. [15] address the problem of learning (deep) decentralized policies with a centralized critic during learning in structured environments. While they do not address the problem of generalization, nor the problem of learning a centralized controller, we use their idea of a separate critic computed based on the full state information during training.

## 3 Multi-agent Task Assignment

We formalize a general MAC problem. To keep notation simple, we present a fixed-size description, but the end goal is to design policies that can be applied to environments of arbitrary size.

As customary in reinforcement learning, the objective of solving the tasks is encoded through a reward function that needs to be maximized over the long run by the coordinated actions of all agents. An environment with $m$ tasks and $n$ agents is modeled as an MDP $\langle \mathcal{S}^m, \mathcal{X}^n, \mathcal{A}^n, r, p \rangle$, where $\mathcal{S}$ is the set of possible states of each *task* (indexed by $j = 1, \ldots, m$), $\mathcal{X}$ and $\mathcal{A}$ are the the set of states and actions of each agent (indexed by $i = 1, \ldots, n$). We denote the joint states/actions by $\mathbf{s} \in \mathcal{S}^m$, $\mathbf{x} \in \mathcal{X}^n$, and $\mathbf{a} \in \mathcal{A}^n$. The reward function is defined as $r : \mathcal{S}^m \times \mathcal{X}^n \times \mathcal{A}^n \to \mathbb{R}$ and the stochastic dynamics is $p : \mathcal{S}^m \times \mathcal{X}^n \times \mathcal{A}^n \to \Delta(\mathcal{S}^m \times \mathcal{X}^m)$, where $\Delta$ is the probability simplex over the (next) joint state set. A joint deterministic policy is defined as a mapping $\pi : \mathcal{S}^m \times \mathcal{X}^n \to \mathcal{A}^n$. We consider the episodic discounted setting where the action-value function is defined as $Q^\pi(\mathbf{s}, \mathbf{x}, \mathbf{a}) = \mathbb{E}_\pi \big[ r(\mathbf{s}, \mathbf{x}, \mathbf{a}) + \sum_{t=1}^T \gamma^t r(\mathbf{s}_t, \mathbf{x}_t, \mathbf{a}_t) \big]$, where $\gamma \in [0, 1)$, $\mathbf{a}_t = \pi(\mathbf{s}_t, \mathbf{x}_t)$ for all $t \geq 1$, $\mathbf{s}_t$ and $\mathbf{x}_t$ are sampled from $p$, and $T$ is the time by when all tasks have been solved. The goal is to learn a policy $\pi$ close to the optimal $\pi^* = \arg\max_\pi Q^\pi$ that we can easily generalize to larger environments.

**Task decomposition.** Following a similar task decomposition approach as [27] and [20], we consider hierarchical policies that first assign each agent to a task, and where actions are given by a lower-level policy that only depends on the state of individual agents and the task they are assigned to. Denoting by $\mathcal{B} = \{ \beta \in \{0, 1\}^{n \times m} : \sum_{j=1}^m \beta_{ij} = 1 \}$ the set of assignment matrices of agents to tasks, an assignment policy first chooses $\hat{\beta}(\mathbf{s}, \mathbf{x}) \in \mathcal{B}$. In the second step, the action for each agent is chosen according to a lower-level policy $\tilde{\pi}$. Using $\pi_i(\mathbf{s}, \mathbf{x})$ to denote the action of agent $i$ and $\hat{\beta}_i(\mathbf{s}, \mathbf{x}) \in \{1, ..., m\}$ for the task assigned to agent $i$, we have $\pi_i(\mathbf{s}, \mathbf{x}) = \tilde{\pi}(s_{\hat{\beta}_i(\mathbf{s}, \mathbf{x})}, x_i)$, where $s_j$ and $x_i$ are respectively the internal states of task $j$ and agent $i$ in the full state $(\mathbf{s}, \mathbf{x})$. In the following, we focus on learning high-level assignment policies responsible for the collaborative behavior, while we assume that the lower-level policy $\tilde{\pi}$ is *known and fixed*.

## 4 A Structured Prediction Approach

In this section we introduce a novel method for centralized coordination. We propose a structured prediction approach in which the agent-task assignment is chosen by solving an optimization problem. Our method is composed of two components: a **coordination inference procedure**, which defines the shape of the optimization problem and thus the type of coordination between agents and tasks, and a **scoring model**, which receives as input the state of agents and tasks and returns the parameters of the objective function of the optimization. The combination of these two components defines an agent-task assignment policy $\hat{\beta}$ that is then passed to the low-level policy $\tilde{\pi}$ (that we assume fixed) which returns the actual actions executed by the agents. Finally, we use a **learning algorithm** to learn the parameters of the scoring model itself in order to maximize the performance of $\hat{\beta}$. The overall scheme of this method is illustrated in Fig. 1.

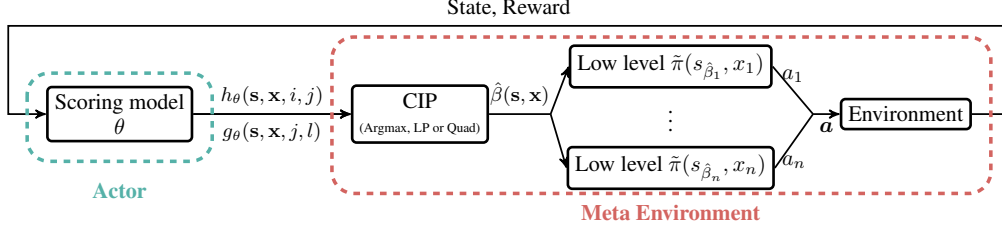

Figure 1: Illustration of the approach, where the agent-task assignment is computed by a coordination inference procedure (CIP) which receives as input agent-task ($h$) and task-task ($g$) scores computed by a scoring model parametrized by $\theta$. The assignment $\hat{\beta}$ is then passed to fixed low level policies that return the actions played by each agent. The learning algorithm tunes $\theta$ and performs "meta-actions" $h_\theta$ and $g_\theta$ on to the "meta-environment" composed by the inference procedure, the low-level policies, and the actual environment.

## 4.1 Coordination Inference Procedures

The collaborative behaviors that we can represent are tied to the specific form of the objective function and its constraints. The formulations we propose are motivated by collaboration patterns important for long-term performance, such as creating subgroups of agents, or spreading agents across tasks.

**Greedy assignment.** The simplest form of assignment is to give a score to each agent-task pair and then assign each agent to the task with the highest score, ignoring other agents at inference time. In this approach, that we refer to as AMAX strategy, a model $h_\theta(\mathbf{x}, \mathbf{s}, i, j) \in \mathbb{R}$ parameterized by $\theta$ receives as input the full state and returns the score of agent $i$ for task $j$. The associated policy is then

$$\hat{\beta}^{\text{AMAX}}(\mathbf{s}, \mathbf{x}, \theta) = \arg\max_{\beta \in \mathcal{B}} \sum_{i,j} \beta_{i,j} h_\theta(\mathbf{s}, \mathbf{x}, i, j), \tag{1}$$

which corresponds to assigning each agent $i$ to the task $j$ with largest score $h_\theta(\mathbf{x}, \mathbf{s}, i, j)$. As a result, the complexity of coordination is reduced from $O(m^n)$ (i.e., considering all possible agent-to-task assignments) down to a linear complexity $O(nm)$ (once the function $h_\theta$ has been evaluated on the full state). We also notice that AMAX bears strong resemblance to the strategy used in [20], where the scores are replaced by approximate value functions computed for any agent-task pair.[2]

**Linear Program assignment.** Since AMAX ignores interactions between agents, it tends to perform poorly in scenarios where a task has a high score for all agents (i.e., $h(s, x, i, j)$ is large for a given $j$ and for all $i$). In this situation, all agents are assigned to the same task, implicitly assuming that the "value" of solving a task is additive in the number of agents assigned to it (i.e., if $n$ agents are assigned to the same task then we could collect a reward $n$ times larger). While this may be the case when the number of agents assigned to the same task is small, in many practical scenarios this effect tends to saturate as more agents are assigned to a single task. A simple way to overcome this undesirable behavior is to impose a restriction on the number of agents assigned to a task. We can formalize this intuition by introducing $\mu_{(i,j)}(\mathbf{s}, \mathbf{x})$ as the *contribution* of an agent $i$ to a given task $j$, and $u_j(\mathbf{s}, \mathbf{x})$ as the *capacity* of the task $j$. In the simplest case, we may know the maximum number of agents $n_j$ that is necessary to solve each task $j$, and we can set the capacity of each task to be $n_j$, and all the contributions $\mu_{i,j}$ to be 1. Depending on the problem, the capacities and contributions are either prior knowledge or learned as a function of the state. Formally, denoting by $\overline{\mathcal{B}}(\mathbf{s}, \mathbf{x})$ the constrained assignment space

$$\overline{\mathcal{B}}(\mathbf{s}, \mathbf{x}) = \left\{ \beta \in \{0,1\}^{n \times m} \middle| \; \forall i, \sum_{j=1}^{m} \beta_{i,j} \leq 1; \forall j, \; \sum_{i=1}^{n} \mu_{i,j}(\mathbf{s}, \mathbf{x})\beta_{i,j} \leq u_j(\mathbf{s}, \mathbf{x}) \right\}, \tag{2}$$

the resulting policy infers the assignment by solving an integer linear program

$$\hat{\beta}^{\text{LP}}(\mathbf{s}, \mathbf{x}, \theta) = \arg\max_{\beta \in \overline{\mathcal{B}}(\mathbf{s}, \mathbf{x})} \sum_{i,j} \beta_{i,j} h_\theta(\mathbf{s}, \mathbf{x}, i, j), \tag{3}$$

Notice that even with the additional constraints in (2), some agents may not be assigned to any task, hence inequality $\sum_{j=1}^{m} \beta_{i,j} \leq 1$ instead of strict equality.

In order to optimize (3) efficiently, we trade off accuracy for speed by solving its linear relaxation using an efficient LP library [1], and retrieving a valid assignment using greedy rounding: Let us denote as $\beta_{i,j}^*$ the solution of the relaxed ILP; we iterate over agents $i$ in descending order of $\max_j \beta_{i,j}^*$, and assign each agent to the task of maximum score that is not already saturated.

**Quadratic Program assignment.** The linear program above avoids straightforward drawbacks of a greedy assignment policy, but is unable to represent grouping patterns that are important on the long-run in coordination and collaboration problems. For instance, it may be convenient to "spread" agents among unrelated tasks, or, on the contrary, group agents together on a single task (up to the constraints) and then move to other tasks in a sequential fashion. Such grouping patterns can be well represented with a quadratic objective function of the form

$$\hat{\beta}^{\text{QUAD}}(\mathbf{s}, \mathbf{x}, \theta) = \operatorname*{arg\,max}_{\beta \in \overline{\mathcal{B}}(\mathbf{s}, \mathbf{x})} \left[ \sum_{i,j} \beta_{i,j} h_\theta(\mathbf{s}, \mathbf{x}, i, j) + \sum_{i,j,k,l} \beta_{i,j} \beta_{k,l} g_\theta(\mathbf{s}, \mathbf{x}, j, l) \right], \qquad (4)$$

where $g_\theta(\mathbf{x}, \mathbf{s}, j, l)$ plays the role of a (signed) distance between two tasks and $\overline{\mathcal{B}}$ is the same set of constraints as in (2). In the extreme case where $g_\theta(\mathbf{x}, \mathbf{s}, ., .)$ is a diagonal matrix, the quadratic part of the objective favors agents to carry on the same task (if the diagonal terms are positive) or on the contrary carry on different tasks (if the terms are negative). In general, negative $g_\theta(\mathbf{x}, \mathbf{s}, j, l)$ disfavors agents to be assigned to $j$ and $l$ at the same time step depending on $|g_\theta(\mathbf{x}, \mathbf{s}, j, l)|$. For instance, in the search and rescue problem, this captures the idea that agents should spread to explore the map.

As for the LP , we optimize a continuous relaxation of (4) using the same rounding procedure. The objective function may not be concave, because there is no reason for $g_\theta(\mathbf{x}, \mathbf{s}, ., .)$ to be negative semi-definite. In practice, we use the Frank-Wolfe algorithm [9] to deal with the linear constraints; the algorithm is guaranteed to converge to a local maximum and was efficient in our experiments.

## 4.2 Scoring Models

In order to allow generalizing the coordination policy $\widehat{\beta}$ to instances of different size, the $h_\theta$ and $g_\theta$ functions should be able to compute scores for pairs agents/tasks and tasks/tasks, independently of the actual amount of those. In order to make the presentation concrete, in the following we illustrate different scoring models in the case where the agents and tasks are objects located in a fixed-size 2D grid, and are characterized by an internal state. The position on the grid is part of this internal state.[3]

**Direct Model (DM).** The first option is to use a fully decomposable approach (*direct model*), where the score for the pair $(i, j)$ only depends on the internal states of agent $i$ and task $j$: $h_\theta(\mathbf{s}, \mathbf{x}, i, j) = \tilde{h}_\theta(s_j, x_i)$ for some function $\tilde{h} : \mathcal{S} \times \mathcal{X} \to \mathbb{R}$. This model only uses the features of the pair of objects to compute the score. Precisely, $\tilde{h}_\theta(s_j, x_i)$ is obtained by concatenating the feature vectors of agent $i$ and task $j$, and by feeding them to a fully-connected network of moderate depth. In the quadratic program strategy, the function $g_\theta$ follows the same structure as $h$ (but uses different weights).

While this approach is computationally efficient, if used in the simple AMAX procedure (1), it leads to a policy that ignores interactions between agents altogether and is thus unable to represent effective collaboration patterns. As a result, the direct model should be paired with more sophisticated inference procedures to achieve more complex coordination patterns. On the other hand, as it computes scores by ignoring surrounding agents and tasks, once learned on small instances, it can be directly applied (i.e., zero-shot generalization) to larger instances independently of the number of agents and tasks.

**General Model.** An alternative approach is to take $h_\theta$ as a highly expressive function of the full state. The main challenge is this case is to define an architecture that can output scores for a variable number of agents and tasks. In the case where agents/tasks are in a 2D grid, we can define a **positional embedding model (PEM)** (see App. **??** for more details) that computes scores following ideas similar to non-local networks [29]. We use a deep convolutional neural network that outputs $k$ features planes at the same resolution as the input. This implies that each cell is associated with $k$ values that we treat as an embedding of the position. We divide this embedding

in two sub-embeddings of size $k/2$, to account for the two kinds of entities: the first $k/2$ values represent an embedding of an agent, and the remaining ones represent an embedding of a task. To compute the score between two entities, we concatenate the embeddings of both entities and the input features of both of them, and run that through a fully connected model, using the same topology as described for the direct model.

By leveraging the *full* state, this model can capture non-local interactions between agents and tasks (unlike the direct model) depending on the receptive field of the convolutional network. Larger receptive fields allow the model to learn more sophisticated scoring functions and thus better policies. Furthermore, it can be applied to variable number of agents and tasks as a position contains at most one agent and one task. Nonetheless, as it depends on the full state, the application to larger instances means that the model may be tested on data points outside the support of the training distribution. As a result, the scores computed on larger instances may be not accurate, thus leading to policies that can hardly generalize to more complex instances.

### 4.3 Learning Algorithm

As illustrated in Fig. 1, the learning algorithm optimizes a policy parametrized by $\theta$ that returns as actions the scores $h_\theta$ and $g_\theta$, while the combination of the assignment $\widehat{\beta}$ returned by the optimization, the low-level policy $\widetilde{\pi}$ and the environment, plays the role of a "meta-environment". While any policy gradient algorithm could be used to optimize $\theta$, in the experiments we use a synchronous Advantage-Actor-Critic algorithm [18], which requires computing a state-value function. As advocated by Rashid et al. [21] in the context of learning decentralized policies, we use a global value function that takes the whole state as input. We use a CNN similar to the one of PEM, followed by a spatial pooling and a linear layer to output the value. This value function is used only during training, hence its parametrization does not impact the potential generalization of the policy (more details in App. **??**).

**Correlated exploration.** Reinforcement learning requires some form of exploration scheme. Many algorithms using decompositional approaches for MAC problems [10, 27, 20, 21] rely on variants of Q-learning or SARSA and directly randomize the low-level actions taken by the agents. However, this approach is not applicable to our framework. In our case, the randomization is applied to the scores (denoted as $H_\theta(\mathbf{s}, \mathbf{x}, i, j)$ and $G_\theta(\mathbf{s}, \mathbf{x}, j, l)$) before passing them to the inference procedure. We can't use a simple gaussian noise, since at the beginning of the training, when the scoring model is random, it would cause the agents to be assigned to different tasks at each step, thus preventing them from solving any task and getting any positive reward. To alleviate this problem, we correlate temporally the consecutive realizations of $H_\theta$ and $G_\theta$ using auto-correlated noise as studied in [e.g., 30, 14], so that the actual sequence of assignments executed by the agent is also correlated. To correlate the parameters over $p$ steps, at time $t$, we sample $H_{t,\theta}(i,j)$ according to (dropping dependence on $(\mathbf{s}_t, \mathbf{x}_t)$ for clarity): $\mathcal{N}\big(h_{t,\theta}(i,j) + \sum_{t'=t-p}^{t-1}(H_{t',\theta}(i,j) - h_{t',\theta}(i,j)), \frac{\sigma}{p}\big)$. This is equivalent to correlating the sampling noise over a sliding window of size $p$. During the update of the model, we ignore the correlation, and assume that the actions were sampled according to $\mathcal{N}(h_{t,\theta}(i,j), \sigma)$.

## 5 Experiments

We report results in two different problems: search and rescue and target selection in StarCraft. Both experiments are designed to test the generalization performance of our method: we learn the scoring models on small instances and the learned policy is tested on larger instances with no additional re-training. We test different combinations of coordination inference procedures and scoring models. Among the inference procedures, AMAX should be considered as a basic baseline, while we expect LP to express some interesting coordination patterns. The QUAD is expected to achieve the better performance in the training instance, although its more complex coordination patterns may not effectively generalize well to larger instances. Among the scoring models, PEM should be able to capture dependencies between agents and tasks in a single instance but may fail to generalize when tested on instances with a number of agents and tasks not seen at training time. On the other hand, the simpler DM should generalize better if paired with a good coordination inference procedure.

The PEM + AMAX combination roughly corresponds to independent A2C learning and can be seen as the standard approach baseline, and we also provied strong hand-crafted baselines. Most previous approaches didn't aim achieving effective generalization, and often relied on fixed-size action spaces, rendering direct comparison impractical.

Table 1: *Search and Rescue*. Average number of steps to solve the validation episodes, depending on the train scenario. $\Delta$ denotes the improvement over baseline. Best results are in bold, with an asterisk when they are statistically ($p < 0.0001$) better than the second best. Results like "*10.3*(1.1%)" mean that the evaluation failed in 1.1% of the test scenarios, and had an average score of 10.3 on the remaining 98.9%. In case of evaluation failures, the reported improvement over baseline are indicative (reported in italics between parenthesis).

| | Train ($n \times m$) | Test | Baseline | Topline | AMAX-PEM | LP-PEM | QUAD-PEM | AMAX-DM | LP-DM | QUAD-DM |
|---|---|---|---|---|---|---|---|---|---|---|
| lower | $2 \times 4$ | $2 \times 4$ | 14.34 | 10.28 | 11.98 | 12.09 | **11.44** | 13.78 | 11.98 | 11.55 |
| is | | $5 \times 10$ | 13.61 | 7.19 | 13.36 | 10.69 | 9.67 | 12.49 | 10.24 | **9.32**\* |
| better | | $8 \times 15$ | 11.8 | n.a | *15.8*(0.7%) | 9.86 | *10.3*(1.1%) | 11.06 | 9.71 | **7.85**\* |
| lower | $5 \times 10$ | $2 \times 4$ | 14.34 | 10.28 | 12.05 | 12.94 | *13.23*(1%) | 13.84 | 12.22 | **11.78**\* |
| is | | $5 \times 10$ | 13.61 | 7.19 | 9.84 | 10.24 | 10.43 | 12.26 | 10.12 | **9.36**\* |
| better | | $8 \times 15$ | 11.8 | n.a | 8.60 | 9.37 | 9.51 | 10.57 | 8.63 | **7.95**\* |
| higher | In domain $\Delta$ | | | | 22% | 20% | 21% | 7% | 21% | **25%** |
| is | Out of domain $\Delta$ | | | | (22%) | 17% | (18%) | 7% | 21% | **29%** |
| better | Total $\Delta$ | | 0% | 38% | (18%) | 18% | (19%) | 7% | 21% | **28%** |

## 5.1 Search and Rescue

**Setting.** We consider a search and rescue problem on a grid environment of 16 by 16 cells. Each instance is characterized by a set of $n$ ambulances (i.e., agents) and $m$ victims (i.e., tasks). The goal is that all the victims are picked up by one of the ambulances as quickly as possible. This problem can be seen as a Multi-vehicle Routing Problem (MVR), which makes it NP-hard.

The reward is $-0.01$ per time-step until the end of an episode (when all the victims have been picked up). The learning task is challenging because the reward is uninformative and coupled; it is difficult for an agent to assign credit to the solution of an individual tasks (i.e., picking up a victim). The assignment policy $\widehat{\beta}$ matches ambulances to victims, while the low-level policy $\widetilde{\pi}$ takes an action to reduce the distance between the ambulance and its assigned victim. In this environment, only one ambulance is needed to pick-up a particular victim, hence the saturation $u_j(\mathbf{s}, \mathbf{x})$ is set to 1. We trained our models on two instances ($n = 2, m = 4$ and $n = 5, m = 10$) and we test it on the trained scenarios, as well as in instances with a larger number of victims and ambulances. At test time, we evaluate the policies on a fixed set of 1000 random episodes (with different starting positions). The agents use the same variance and number of correlated steps as they had during training. The results are summarized in Tab. 1, where we report the average number of steps required to complete the episodes. We also report the results of a greedy baseline policy that always assigns each ambulance to the closest victim, and a topline policy that solves each instance optimally (see App. **??** for more details). Because of its computational cost, the topline for the biggest instance ($8 \times 15$) is not available. In the last rows of the table, we aggregate the average improvements over the baseline ($100 * \frac{\text{baseline} - \text{method}}{\text{baseline}}$). The in-domain scores correspond to the scores obtained when the test instance matches the train instance. Conversely, the out of domain scores correspond to the performances on unseen instances. Note that no model was trained on $8 \times 15$.

**Results.** Firstly, the PEM scoring model tends to overfit to the train scenario, leading to poor generalization (i.e., in some configuration it fails to solve the problem). On the other hand, for the DM, the generalization is very stable. Regarding the inference procedures [4], AMAX tends to perform at least as well as the greedy baseline, by learning how to compute the relevant distance function between an ambulance and a victim. The LP strategy can rely on the same distance function and perform better, by since it enforces the coordination and avoids sending more than one ambulance to the same victim. Finally, the QUAD strategy is able to learn long-term strategies, and in particular how to spread efficiently the ambulances across the map (e.g., if two victims are very close, it is wasteful to assign two distinct ambulances to them, since one can efficiently pick-up both victims sequentially, while the other ambulance deals with further victims) (see App.**??** for more discussion).

## 5.2 Target Selection in StarCraft

**Setting.** We focus on a specific sub-problem in StarCraft: battles between two groups of units. This setting, often referred to as *micromanagement*, has already been studied in the literature, using a mixture of scripted behaviours and search [4, 3, 19, 5], or using RL techniques [28, 8]. In these

battles, a crucial aspect of the policy is to assign a target enemy unit (*the task*) to each of our units (*the agents*), in a coordinated way. Since we focus on the agent-task assignment (the high-level policy $\widehat{\beta}$), we use a simple low-level policy for the agents (neither learnt nor scripted) relying on the built-in "attack" command of the game, which moves each unit towards its target and shoots as soon as the target is in range. This contrasts with previous works, which usually allow more complex movement patterns (e.g., retreating while the weapon is reloading). While such low-level policies could be integrated in our framework, we preferred to use the simplest "attack" policy to better assess the impact of the high-level coordination.

In this problem, the capacity $u_j(\mathbf{s}, \mathbf{x})$ of a task $j$ is defined as the remaining health of the enemy unit, and the contribution $\mu_{i,j}(\mathbf{s}, \mathbf{x})$ of an agent $i$ to this task is defined as the amount of damage dealt by unit $i$ to the enemy $j$. These constraints are meant to avoid dealing more damage to an enemy than necessary to kill it, a phenomenon known as *over-killing*.

Given the poor results of PEM in the previous experiment, we only train DM with all the possible inference procedures. Each unit is represented by its features: whether it is an enemy, its position, velocity, current health, range, cool-down (number of frames before the next possible attack), and one hot encoding of its type. This amounts to 8 to 10 features per units, depending on the scenario. For training, we sample 100 sets of hyper-parameters for each combination of model/scenario, and train them for 8 hours on a Nvidia Volta. In this experiment, we found that the training algorithm is relatively sensitive to the random seed. To better asses the performances, we re-trained the best set of hyper-parameters for each model/scenario on 10 random seeds, for 18 hours. The performances we report are the median of the performances of all the seeds, to alleviate the effects of the outliers. The results are aggregated in Tab. 2. Although the number of units is a good indicator of the difficulty of the environment, whether the numbers of units are balanced in both teams dramatically change the "dynamics" of the game. For instance, zh10v12 is unbalanced and thus much more difficult than zh11v11, which is balanced. The performance of the baseline can be seen as a relatively accurate estimate of the difficulty of the scenario. See App. **??** for a description of the heuristics used for comparison, and App. **??** for a description of the training scenarios. We also provide a more detailed results in App. **??**.

**Results.** As StarCraft is a real-time game, one first concern regards the runtime of our algorithm. In the biggest experiment, involving 80 units vs 82, our algorithm returned actions in slightly more than 500ms (5ms for the forward in the model, 500ms to solve the inference of QUAD ). Given the frequency at which we take actions (every 6 frames), such timings allow real-time play in StarCraft.

Amongst the scenarios, the Wraith setting (w$N$v$M$) are the ones where the assumption of independence between the tasks holds the best, since in this case there are no collisions between units. These scenarios also require good coordination, since it is important to focus fire on the same unit. During these battles, both armies tend to overlap totally, hence it becomes almost impossible to use surrogate coordination principles such as targeting the closest unit. In this case, the quadratic part of the score function is crucial to learn focus-firing and results show that without the ability to represent such a long-term coordination pattern, both LP and AMAX fail to reach the same level of performance. Notably, the coordination pattern learned by QUAD generalizes well, outperforming the best heuristics in instances as much as 5 times the size of the training instance.

Table 2: *StarCraft*. Average win-rate of different methods (best in bold). See confidence intervals in the full table in App **??**

| Train | Test | Best heuristic | LP | QUAD | AMAX |
|---|---|---|---|---|---|
| m10v10 | m5v5 | 0.88 | **0.90** | 0.83 | 0.84 |
| | m10v10 | 0.77 | **0.94** | 0.83 | 0.82 |
| | m10v11 | 0.25 | **0.52** | 0.28 | 0.29 |
| | m15v15 | 0.75 | **0.92** | 0.69 | 0.77 |
| | m15v16 | 0.40 | **0.68** | 0.32 | 0.43 |
| | m30v30 | 0.69 | **0.74** | 0.06 | 0.36 |
| w15v17 | w15v17 | 0.81 | 0.53 | **0.89** | 0.30 |
| | w30v34 | 0.90 | 0.76 | **0.99** | 0.37 |
| | w30v35 | 0.60 | 0.56 | **0.94** | 0.24 |
| | w60v67 | 0.07 | 0.33 | **0.72** | 0.13 |
| | w60v68 | 0.01 | 0.21 | **0.52** | 0.07 |
| | w80v82 | 0.32 | 0.11 | **0.36** | 0.03 |
| zh10v10 | zh10v10 | 0.86 | **0.90** | 0.83 | 0.84 |
| | zh10v11 | 0.30 | **0.46** | 0.24 | 0.40 |
| | zh10v12 | 0.03 | **0.06** | 0.01 | **0.06** |
| | zh11v11 | **0.87** | **0.87** | 0.75 | 0.80 |
| | zh12v12 | **0.85** | 0.82 | 0.64 | 0.75 |

The other setting, Marine (m$N$v$M$) and Zergling-Hydralisk (zh$N$v$M$) break the independence assumption because the units now have collisions. It is even worse for the Zerglings, since they are melee units. The coordination patterns are then harder to learn for the QUAD model, and they generalize poorly. However, these scenarios with collisions also tend to require less long-term

coordination, and the immediate coordination patterns learned by the LP model are enough to significantly outperform the heuristics, even when transferring to unseen instances.

## 6 Conclusion

In this paper we proposed a structured approach to multi-agent coordination. Unlike previous work, it uses an optimization procedure to compute the assignment of agents to tasks and define suitable coordination patterns. The parameterization of this optimization procedure is seen as the continuous output of an RL trained model. We showed on two challenging problems the effectiveness of this method, in particular in generalizing from small to large instance.

## Footnotes

[1]StarCraft and its expansion StarCraft: Brood War are trademarks of Blizzard Entertainment[TM].

[2]An alternative approach is to sample assignments proportionally to $h_\theta(\mathbf{s}, \mathbf{x}, i, j)$. Preliminary empirical tests of this procedure performed worse than AMAX and thus we do not report its results.

[3]Notice that the direct model illustrated below does not leverage this specific scenario, which, on the other hand, is needed to define the general model.

[4] We study their performance when paired with DM.

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
