[Supplementary Material 1 · Appendix.pdf]



Figure 2: The PEM model for 3 agents and 2 tasks, arrows of interactions are only drawn for 2 agents each with one task, and one interaction of the two tasks. Agents are in cold (blue/green) colors, tasks are in warm (red/orange) colors.

## A  The Positional Embedding Model

A more complete illustration of the PEM model is reported in Fig. 2.

---

**Algorithm 1** Structured Prediction RL algorithm - Worker thread

---

*// $p$ is the number of correlated steps, $\sigma$ the exploration standard deviation*
*// We denote by $Queue_p$ a queue of fixed size $p$. We assume we can sum all the elements in it.*
$T \leftarrow 0$
**repeat**
   Start new a episode, $t \leftarrow 0$
   **for all** agent $i$ and task $j$ **do**
      Init noise history for the linear part to 0: $noise\_hist\_lin(i,j) = \text{Queue}_p(0)$
   **end for**
   **for all** pair of task $j$, $k$ **do**
      Init noise history for the quadratic part to 0: $noise\_hist\_quad(j,k) = \text{Queue}_p(0)$
   **end for**
   **repeat**
      Observe state $s_t$
      **for all** agent $i$ and task $j$ **do**
         Compute $h_\theta(i,j,s_t)$ using the network
         Sample the actual action $H_{i,j}(s_t) \sim \mathcal{N}\left(h_\theta(i,j,s_t) + \sum noise\_hist\_lin(i,j), \frac{\sigma}{p}\right)$
         Store the current noise $noise\_hist\_lin(i,j)$.append($H_{i,j}(s_t) - h_\theta(i,j,s_t)$)
         Compute $\mu_{i,j}(s_t)$ the contribution of the agent to the task
         Compute $u_j(s_t)$ the capacity of the task
      **end for**
      **for all** pair of task $j$, $k$ **do**
         Compute $g_\theta(j,k,s_t)$ using the network
         Sample the actual action $G_{j,k}(s_t) \sim \mathcal{N}\left(g_\theta(j,k,s_t) + \sum noise\_hist\_quad(j,k), \frac{\sigma}{p}\right)$
         Store the current noise $noise\_hist\_quad(j,k)$.append($G_{j,k}(s_t) - g_\theta(j,k,s_t)$)
      **end for**
      Compute the assignment using the constrained optimizer $\beta \leftarrow \text{solve}(H, G, \mu, u)$
      Execute the assignment in the environment, observe reward $r_t$
      The policy is $\pi(s_t) = [h_\theta(s_t), g_\theta(s_t)]$, and the action $a(s_t) = [H, G]$
      Send to the learner thread $(\pi(s_t), a(s_t), s_t, r_t)$
      $t \leftarrow t+1$
      $T \leftarrow T+1$
   **until** Episode ends
**until** $T > T_{max}$

---

---
**Algorithm 2** Structured Prediction RL algorithm - Learner thread
---

*// $p$ is the number of correlated steps, $N$ is the return length we use, $\sigma$ the exploration standard deviation*
*$\gamma$ is the discount factor, $\lambda$ is the policy weight*
**repeat**
    Reset gradients $d\theta \leftarrow 0$
    Receive $N$ consecutive samples from a worker thread $[(\pi_1, a_1, s_1, r_1), \dots, (\pi_N, a_N, s_N, r_N)]$
    $R = \begin{cases} 0 & \text{if } s_N \text{ is terminal} \\ V(s_N, \theta) & \text{otherwise} \end{cases}$
    **for** $t = N-1$ **to** $1$ **do**
        $R = \begin{cases} r_t & \text{if } s_t \text{ is terminal} \\ r_t + \gamma R & \text{otherwise} \end{cases}$
        Accumulate value loss $d\theta \leftarrow d\theta + \nabla_\theta \left( |R - V(s_t, \theta)| \right)$
        Compute advantage $A_t \leftarrow R - V(s_t)$
        Compute new policy $\pi_{\text{new},t}(s_t, \theta)$ using the network
        Compute likelihood of action according to old policy (assuming $a_t \sim \mathcal{N}(\pi_t, \sigma)$) $l_{\text{old}} \leftarrow$ NormalPDF$(a_t, \pi_t, \sigma)$
        Compute likelihood according to new policy (assuming $a_t \sim \mathcal{N}(\pi_{\text{new},t}, \sigma)$) $l_{\text{new}} \leftarrow$ NormalPDF$(a_t, \pi_{\text{new},t}, \sigma)$
        Compute importance ratio $ir \leftarrow \frac{l_{\text{new}}}{l_{\text{old}}}$
        Accumulate policy loss $d\theta \leftarrow d\theta + \lambda \nabla_\theta (ir * A * \log(l_{\text{new}}))$
    **end for**
    Take an optimization step according to $d\theta$.
**until** training ends

---

## B    Formal description of the training algorithm

In the experiments of this paper, we chose to build upon A2C with continuous actions, but in principle, any continuous policy-gradient algorithm could be used. The pseudo-code of the learning algorithm is presented in Algorithm 1 for the worker threads and Algorithm 2 for the learning threads.

## C    Search and rescue Experimental Setup

Figure 3: Search and rescue environment, with 5 ambulances and 10 victims.

### C.1    Details of the Problem

We consider a search and rescue problem on a 16x16 grid. Each ambulance can move in any of the 8 adjacent cell at each step, and is assumed to be able to pick-up an infinite number of victims. At the beginning of each episode, $m$ victims and $n$ ambulances are spawned uniformly at random on the grid. An ambulance picks up a victim as soon as it reaches its cell, regardless of whether this ambulance was effectively assigned to that victim or if it visited the cell contingently. When it happens, the state of the victim changes to reflect the fact that that task is solved, yet nothing prevents the agent to keep assigning ambulances to it.

## C.2 Training

We optimize the hyper-parameters of the learning procedure using a random search. We sample 128 sets of hyper-parameters for each combination of model/scenario, and we train the models during 8 hours on a Nvidia Volta. We report the performance of the best performing model after training according to the average reward on training problems. The parameters that were tuned are the following: learning rate of the value network (we sample $c \in [0, 5]$, uniformly at random, and use $10^{-c}$), the learning rate of the policy network (same sampling), the variance of the random policy (uniformly in $[0.1, 2]$), the number of correlated steps in the exploration (uniformly in $[1, 10]$), the number of steps of the $n$-step return (uniformly in $[2, 5]$), and the optimization algorithm (SGD or Adam [16]).

## C.3 Optimal algorithm

For the small instances of the problem, it is possible to use an exact algorithm (topline) that solves the Multi-Vehicle Routing Problem exactly. To do it efficiently, we first pre-compute for each subset of victims (there are $2^m$ possible subsets), the shortest path to visit them all, starting from each of the victims of the subset (that is, if a subset contains 5 victims, we compute the 5 shortest paths that go through all of them starting from each of the victims). This pre-computation phase is done using a dynamic programming algorithm that has a complexity of $O(2^m m^3)$. This allows us to compute, for a given ambulance, what is the optimal way of visiting a given subset of the victims: we simply need to choose which victim to visit first and then execute the optimal path that we have cached between the remaining ones. What remains to be done is to partition the victims in $n$ sets, and assign those sets to the $n$ ambulances, so that the maximal time required by the ambulances to visit their subset optimally is minimized. To solve this partition/assignment problem efficiently, we model it as an Integer Linear Program, and solve it exactly using an efficient branch-and-cut solver (SCIP [10])

## C.4 Model

**Features.** The input of the networks consists in 4 feature planes: the first one contains a 1 if the corresponding cell contains a victim (0 otherwise), the second contains a 1 if the corresponding cell contains an ambulance, and the last two planes contain respectively the x/y coordinates of the entity (victim or ambulance) contained in that cell, if there is any.

**Network.** The value network is a residual network ([13]) made of 3 residual blocks of 2 convolutional layers, with kernel size 3, stride 1, and padding such that the output of each layer has the same spatial dimension as the input. Each convolutional layer outputs 32 feature planes, and we use ReLUs as non-linearities, as well as BatchNorm layers for regularization. The output of the last block is connected to a fully connected layer, which outputs one single value, the value function. For the direct model, we use a simple fully connected network, with 3 linear layers separated by ReLUs, with 32 hidden units each. The architecture of the PEM network is the same as the value network.

We train the model using a synchronous Advantage-Actor-Critic algorithm (A2C) (see [20] for the asynchronous version), using the ELF platform [30]. We batch 128 observations together, and run 256 agents in parallel.

## C.5 Detailed results analysis

For the AMAX strategy, the experiments show that the model that has access to the the full view of the map (PEM) performs better on the domain where it was trained, w.r.t. a model that only has access to the features of the objects (DM). This means that the model learns something that is more useful (as a parameterization of the LP /QUAD ) than simply the distance between objects, and probably takes into account some local patterns of the other objects on the map. However, this better performance in-domain comes at a cost of poor generalization: on the $8 \times 15$ scenario, the AMAX PEM model and the QUAD PEM model trained on the $2 \times 4$ gets stuck in a loop in a fraction of the evaluation episodes, for which they are not able to complete the overall problem (in the table we report the performance averaged over non-failing episodes). This is probably because the receptive fields of the network are significantly more saturated (their average activations have a higher value) in the $8 \times 15$ scenario than the models learned in the $2 \times 4$ training scenario. By contrast, the models trained on $5 \times 10$ is not affected by this. The phenomenon also occurs in the opposite direction: the QUAD PEM

Figure 4: (left) Learning curves of the Quad models; (right) The distance computed by model (depicted as the density), is skewed towards the corner, compared to the ground-truth distance (depicted as the contours)

trained on $5 \times 10$ experiences failures on the smaller environment $2 \times 4$, indicating some form of overfitting.

Another interesting observation is that the AMAX PEM agent performs on average 7% better than the baseline. This is unexpected since the only features it has access to are the positions of the ambulance and the victim, thus it would be expected to learn to send each ambulance to the closest victim. We credit this development to the ability of the model to learn to break ties between equally close victims, which it does by choosing the victims that are on the furthest peripheral positions of the map. Since these positions tend to be the outliers, they are on average more difficult to access than the ones in the middle, so it is beneficial to favor them. The baseline breaks ties randomly and thus does not account for that. This can be observed in Fig 4b in which we plotted the similarity function learned by the model, by computing the score the model would give when the agent is located at the red dot and the task is located anywhere else. While the function it generates resembles the ground-truth distance, the distribution is slightly skewed towards the bottom-left corner.

Overall, the constrained strategies (LP and QUAD ) perform better than the AMAX strategy, and the the QUAD strategy is better than the LP strategy, irrespective of the feature model used. This is an evidence that the structure introduced into the inference procedure is indeed able to improve the performance of the agent. These strategies also leverage the structure of the problem through the constraints on the assignment, which means that there is less room for learning something that cannot be computed directly from the pairwise positions using the additional information provided by the access to the full view. Since the extra information does not directly help, it slows down the learning procedure, and hinders the final performance. This can be seen in Fig 4a, where the PEM agent is slower to converge, and converges to a worse performance than the DM agent, when both use the QUAD strategy.

## D  StarCraft experimental setup

All the experiments are played on a fully empty map, where the units are centered in the middle, outside their respective fire range. Because of the kind of movement policy we use, the units never go near the edge of the map, hence there is no collision involved but the collisions between units themselves. The bot takes its actions every 3 frames, but we reevaluate the target assignment only every 6 frames. If the target of a unit dies in the meantime, then it does nothing until the next re-assignment. The reward at time $t$ is defined as

$$r_t = (\text{ourHealth}_t - \text{ourHealth}_{t-1} + \text{theirHealth}_{t-1} - \text{theirHealth}_t)/\text{ourHealth}_0.$$

At the beginning of every battle, the units are spawned according to a seeded normal distribution, which is skewed towards the Y dimension, in order to form more coherent initial spread of the units. Even though the starting positions are deterministic with respect to the seed, the outcome of a battle is not, even if the policy of the players are totally deterministic. This is due to the internal randomness of the game engine, which affects things like the random miss probability of each attack (any attack has a $\frac{1}{256}$ probability of failing) and the initial orientation of the units.

The opponent army is controlled by the built-in AI. In practice, it means that we give an "attack-move" order to the opponent, with the target position being the centroid of our units. This order is repeated every 60 frames with an updated centroid. The attack-move order causes the built-in AI to take over the unit, globally moving it towards the target position and attacking any visible unit along the way.

We wish to emphasize the fact that all the details of the experimental protocol have a significant impact on the outcome of the battles. In particular, the frequency at which the opponent's attack-move command is re-issued matters: if the frequency is too high, then the units tend to attack less, because spamming orders have some weird effects on the game engine. Conversely, if the frequency is too low, then the attack point might be significantly off the centroid of our army, leading to sub-optimal attacking behaviour. Similarly, the initial positions of the units plays an important role. As a result, the exact win-rate are not directly comparable with previous works, where neither precise details of the setup nor source code are available. The differences can be observed even in the win-rate of the baselines heuristics. For this work, we refer to the source code in the supplementary material for the exact details used.

## D.1   Scenarios

We consider three different kinds of scenarios:

**Wraith**   These are ranged flying units, which means that they don't collide with any other unit. We denote these scenarios as w$N$v$M$ where $N$ is the size of our army, and $M$ is the size of the enemy army. Following the previous works, we train on the imbalanced scenario w15v17, where the two additional units given to the opponent are required to make the scenario challenging.

**Marines**   These are ranged ground units, which do have collisions. We denote these scenarios as m$N$v$M$ where $N$ is the size of our army, and $M$ is the size of the enemy army. Since the built-in AI has a better control policy for ground units, we train on the balanced scenario m10v10, which is challenging enough.

**Zergling-Hydralisk**   Zerglings are fast melee units (they can only attack if they are in contact of their target), while Hydralisks are slower ranged units. In this scenario, we investigate the opportunity to learn distinct behaviours depending on the type of the agent or its task. To further amplify their relative capabilities, we give the Zerglings a speed boost ("Metabolic Boost" upgrade), and the Hydralisks a range boost ("Grooved Spines" upgrade). We denote these scenarios as "zh$N$v$M$" which correspond to $N$ Zerglings and $N$ Hydralisks versus $M$ Zerglings and $M$ Hydralisks. We train on zh10v10.

## D.2   Baseline Heuristics

Here is a description of the heuristics used as a comparison:

**Closest (c)**   Each unit independently picks the units that is the closest, as measured by the distance function used by the game engine. Ties are broken using the unit internal ID, which is randomly assigned at the beginning of the game but consistent for the duration of the episode.

**Weakest Closest (wc)**   All units are collectively assigned to the weakest enemy unit. The distance is used as a tie breaker.

**Weakest closest No-Overkill (wcnok)**   . We select the weakest-closest enemy unit, then assign greedily in an arbitrary order as many units as possible as long as the total sum of damage to this unit is lesser than its health. When this enemy is saturated, if some of our units don't have a target yet, then we select the second weakest-closest, and so on until all enemy units have been exhausted or all our units have a target.

**Weakest Closest No-Overkill No Change(wcnoknc)**   Same as wcnok, but once a unit starts attacking a target, it keeps doing so until the target dies. When the target dies, a new target is computed as in wcnok Keeping the same target can reduce some instability in the assignment found by wcnok, but can lead to over-killing.

| Experiment | Model | Learn. rate | Policy-loss weight $\lambda$ | Explor. std-dev $\sigma$ | Returns length | # correlated steps |
|---|---|---|---|---|---|---|
| Marine | QUAD | 4.272e-05 | 1.585e+01 | 2.910e+00 | 5 | 10 |
| | LP | 2.280e-05 | 2.157e-03 | 7.017e-01 | 6 | 5 |
| | AMAX | 4.330e-05 | 5.396e-01 | 2.418e+00 | 2 | 2 |
| Wraith | QUAD | 2.826e-05 | 2.514e+02 | 1.899e+00 | 6 | 7 |
| | LP | 5.600e-05 | 5.534e-01 | 4.244e-01 | 4 | 6 |
| | AMAX | 2.525e-05 | 6.847e+01 | 1.994e+00 | 8 | 10 |
| Zergling-Hydra | QUAD | 5.325e-05 | 6.277e-01 | 7.185e-01 | 3 | 3 |
| | LP | 6.534e-05 | 4.455e-02 | 2.902e+00 | 8 | 1 |
| | AMAX | 1.885e-05 | 2.119e-02 | 1.881e+00 | 3 | 10 |

Table 3: Best hyper-parameters found through random search on the different models

**Weakest Closest No-Overkill Smart (wcnoks)**  Same as wcnoknc, except that the target is kept only as long as it doesn't risk causing over-killing. When it does, a new target is computed as in wcnok.

**Random No change (rand-nc)**  Each unit pick a target at random at the beginning of the episode, and keep attacking it until it is dead. When that happens, a new target is picked randomly.

### D.3  Model

The scoring models $h_\theta$ and $g_\theta$ are fully-connected networks consisting of 3 linear layers with ReLU. To compute $h_\theta(i, j)$, we give as input to the network the concatenation of the features of ally unit $i$ and the features of enemy unit $j$, along with 2 additional features: a boolean flag that indicates whether $i$ was attacking $j$ in the previous step (this is meant to facilitate temporal consistency of the actions), and the distance between both units, as computed by the internal game engine. The input of $g_\theta(j, k)$ only contains the features of enemy units $j$ and $k$, with no additional features. In this experiment, we use 64 agents in parallel during training, and batch 32 observations together. The reward is joint, and consist in the normalized instantaneous delta of health of our units and the enemy units.

### D.4  Hyper-parameters

All models use the same network architecture, using 3 layers of fully-connected with 32 units and ReLUs in-between. The value function uses a residual network made of 5 blocks of 2 convolutional layers, with 16 feature layers and kernel size 3, operating on a 100x100 view of the state. The discount factor $\gamma$ is set to 0.999.

For all the models, we sample randomly the remaining hyper-parameters, as follows: For the learning rate we sample $c$ u.a.r in $[-6, -3]$, and use $10^c$, for the policy-loss weight $\lambda$, we sample $d$ u.a.r in $[-3, 3]$ and use $10^d$, the exploration standard deviation $\sigma$ is sampled u.a.r in $[0.1, 3]$, the return-length used in A2C is an integer sampled u.a.r in $[2, 10]$, and the number of correlated exploration steps is an integer sampled u.a.r. in $[1, 10]$. We report the best parameters found for all model in all experiments in Table 3.

### D.5  Detailed Results

In Table 4, we report the performance of all the heuristics, as well as the confidence intervals. In the Marine and Zergling-Hydralisk scenarios, the "closest" heuristic tends to dominate, suggesting that there is not much value in learning a focus-firing policy. However, in the Wraith scenarios, the best performing heuristic are the more complicated ones that focus fire on the weakest while preventing over-killing. This is the type of policies that only the QUAD model is able to represent, explaining its dominance in this scenario. AMAX and LP manage to do better than the closest heuristic by using enemy positions as a way to reach a consensus: they tend to favor picking targets which have a high $y$ coordinate, for example, so that they all end-up picking the same in the beginning, creating ad-hoc collaboration. However, as the battle goes on, this coordination scheme is not as efficient, since the battle tend to be messy.

| Train | Test | Heuristics | | | | | | RL | | |
|---|---|---|---|---|---|---|---|---|---|---|
| | | c | wc | wcnok | wcnoknc | wcnoks | rand-nc | LP DM | QUAD DM | AMAX |
| m10v10 | m5v5 | 0.77 ± 0.03 | *0.88 ± 0.02* | 0.56 ± 0.03 | 0.86 ± 0.02 | 0.83 ± 0.02 | 0.15 ± 0.02 | **0.90 ± 0.02** | 0.83 ± 0.02 | 0.84 ± 0.02 |
| | m10v10 | *0.77 ± 0.03* | 0.44 ± 0.03 | 0.00 ± 0.00 | 0.45 ± 0.03 | 0.56 ± 0.03 | 0.01 ± 0.01 | **0.94 ± 0.02** | 0.83 ± 0.02 | 0.82 ± 0.02 |
| | m10v11 | *0.25 ± 0.03* | 0.07 ± 0.02 | 0.00 ± 0.00 | 0.05 ± 0.01 | 0.11 ± 0.02 | 0.00 ± 0.00 | **0.52 ± 0.03** | 0.28 ± 0.03 | 0.29 ± 0.03 |
| | m15v15 | *0.75 ± 0.03* | 0.03 ± 0.01 | 0.00 ± 0.00 | 0.14 ± 0.02 | 0.18 ± 0.02 | 0.00 ± 0.00 | **0.92 ± 0.02** | 0.69 ± 0.03 | 0.77 ± 0.03 |
| | m15v16 | *0.40 ± 0.03* | 0.00 ± 0.00 | 0.00 ± 0.00 | 0.03 ± 0.01 | 0.02 ± 0.01 | 0.00 ± 0.00 | **0.68 ± 0.03** | 0.32 ± 0.03 | 0.43 ± 0.03 |
| | m30v30 | *0.69 ± 0.03* | 0.00 ± 0.00 | 0.00 ± 0.00 | 0.00 ± 0.00 | 0.00 ± 0.00 | 0.00 ± 0.00 | **0.74 ± 0.03** | 0.06 ± 0.02 | 0.36 ± 0.03 |
| w15v17 | w15v17 | 0.07 ± 0.02 | 0.00 ± 0.00 | 0.58 ± 0.03 | 0.33 ± 0.03 | *0.81 ± 0.02* | 0.01 ± 0.01 | 0.53 ± 0.03 | **0.89 ± 0.02** | 0.30 ± 0.03 |
| | w30v34 | 0.01 ± 0.01 | 0.00 ± 0.00 | 0.36 ± 0.03 | 0.31 ± 0.03 | *0.90 ± 0.02* | 0.01 ± 0.01 | 0.76 ± 0.03 | **0.99 ± 0.01** | 0.37 ± 0.03 |
| | w30v35 | 0.00 ± 0.00 | 0.00 ± 0.00 | 0.10 ± 0.02 | 0.08 ± 0.02 | *0.60 ± 0.03* | 0.01 ± 0.01 | 0.56 ± 0.03 | **0.94 ± 0.02** | 0.24 ± 0.03 |
| | w60v67 | 0.00 ± 0.00 | 0.00 ± 0.00 | 0.00 ± 0.00 | 0.00 ± 0.00 | *0.07 ± 0.02* | 0.00 ± 0.00 | 0.33 ± 0.03 | **0.72 ± 0.03** | 0.13 ± 0.02 |
| | w60v68 | 0.00 ± 0.00 | 0.00 ± 0.00 | 0.00 ± 0.00 | 0.00 ± 0.00 | *0.01 ± 0.01* | 0.00 ± 0.00 | 0.21 ± 0.03 | **0.52 ± 0.03** | 0.07 ± 0.02 |
| | w80v82 | 0.00 ± 0.00 | 0.00 ± 0.00 | 0.09 ± 0.02 | 0.08 ± 0.02 | *0.32 ± 0.03* | 0.00 ± 0.00 | 0.11 ± 0.02 | **0.36 ± 0.03** | 0.03 ± 0.01 |
| zh10v10 | zh10v10 | *0.86 ± 0.02* | 0.26 ± 0.03 | 0.00 ± 0.00 | 0.54 ± 0.03 | 0.64 ± 0.03 | 0.00 ± 0.00 | **0.90 ± 0.02** | 0.83 ± 0.02 | 0.84 ± 0.02 |
| | zh10v11 | *0.30 ± 0.03* | 0.01 ± 0.01 | 0.00 ± 0.00 | 0.04 ± 0.01 | 0.09 ± 0.02 | 0.00 ± 0.00 | **0.46 ± 0.03** | 0.24 ± 0.03 | 0.40 ± 0.03 |
| | zh10v12 | *0.03 ± 0.01* | 0.00 ± 0.00 | 0.00 ± 0.00 | 0.00 ± 0.00 | 0.00 ± 0.00 | 0.00 ± 0.00 | **0.06 ± 0.02** | 0.01 ± 0.01 | **0.06 ± 0.01** |
| | zh11v11 | ***0.87 ± 0.02*** | 0.15 ± 0.02 | 0.00 ± 0.00 | 0.38 ± 0.03 | 0.56 ± 0.03 | 0.00 ± 0.00 | **0.87 ± 0.02** | 0.75 ± 0.03 | 0.80 ± 0.02 |
| | zh12v12 | ***0.85 ± 0.02*** | 0.05 ± 0.01 | 0.00 ± 0.00 | 0.23 ± 0.03 | 0.42 ± 0.03 | 0.00 ± 0.00 | 0.82 ± 0.02 | 0.64 ± 0.03 | 0.75 ± 0.03 |

Table 4: Results on StarCraft: average win-rate of the different methods and all the heuristics. Bests results are in bold, the best heuristic on each scenario is in italics. We report 95% confidence intervals (using the Normal approximation interval).

In the Marine scenario, all three models usually learn to wait in the beginning of the battle, and start picking a target only when the enemies are about to get in range. This strategy allows them to keep a rather good formation, which they would otherwise last, had they picked a wrong target. This gives them a little edge in the battle, and then they proceed by following a strategy that resemble attacking the closest unit.

In the Zergling-Hydralisk scenarios, the models learn to wait in the beginning as well. This is an exploit on the rushing behaviour of the opponent: the faster zerglings of the enemy will engage first, while its hydralisks are lagging behind. This allows the model to first clear up the zergling wave with the combined forces of their zerglings and hydras, before turning to the enemy hydralisks.

We provide some replay video as a supplementary file.



[Supplementary Material 2]

# `enum` --- support for enumerations

An enumeration is a set of symbolic names (members) bound to unique, constant values. Within an enumeration, the members can be compared by identity, and the enumeration itself can be iterated over.

## Module Contents

This module defines two enumeration classes that can be used to define unique sets of names and values: `Enum` and `IntEnum`. It also defines one decorator, `unique`.

`Enum`

Base class for creating enumerated constants. See section Functional API for an alternate construction syntax.

`IntEnum`

Base class for creating enumerated constants that are also subclasses of `int`.

`unique`

Enum class decorator that ensures only one name is bound to any one value.

## Creating an Enum

Enumerations are created using the `class` syntax, which makes them easy to read and write. An alternative creation method is described in Functional API. To define an enumeration, subclass `Enum` as follows:

```
>>> from enum import Enum
>>> class Color(Enum):
...     red = 1
...     green = 2
...     blue = 3
```

Note: Nomenclature

- The class `Color` is an *enumeration* (or *enum*)
- The attributes `Color.red`, `Color.green`, etc., are *enumeration members* (or *enum members*).
- The enum members have *names* and *values* (the name of `Color.red` is `red`, the value of `Color.blue` is 3, etc.)

Note:

Even though we use the `class` syntax to create Enums, Enums are not normal Python classes. See How are Enums different? for more details.

Enumeration members have human readable string representations:

```
>>> print(Color.red)
Color.red
```

...while their `repr` has more information:

```
>>> print(repr(Color.red))
<Color.red: 1>
```

The *type* of an enumeration member is the enumeration it belongs to:

```
>>> type(Color.red)
<enum 'Color'>
>>> isinstance(Color.green, Color)
True
>>>
```

Enum members also have a property that contains just their item name:

```
>>> print(Color.red.name)
red
```

Enumerations support iteration. In Python 3.x definition order is used; in Python 2.x the definition order is not available, but class attribute __order__ is supported; otherwise, value order is used:

```
>>> class Shake(Enum):
...     __order__ = 'vanilla chocolate cookies mint'  # only needed in 2.x
...     vanilla = 7
...     chocolate = 4
...     cookies = 9
...     mint = 3
...
>>> for shake in Shake:
...     print(shake)
...
Shake.vanilla
Shake.chocolate
Shake.cookies
Shake.mint
```

The __order__ attribute is always removed, and in 3.x it is also ignored (order is definition order); however, in the stdlib version it will be ignored but not removed.

Enumeration members are hashable, so they can be used in dictionaries and sets:

```
>>> apples = {}
>>> apples[Color.red] = 'red delicious'
>>> apples[Color.green] = 'granny smith'
>>> apples == {Color.red: 'red delicious', Color.green: 'granny smith'}
True
```

# Programmatic access to enumeration members and their attributes

Sometimes it's useful to access members in enumerations programmatically (i.e. situations where Color.red won't do because the exact color is not known at program-writing time). Enum allows such access:

```
>>> Color(1)
<Color.red: 1>
>>> Color(3)
<Color.blue: 3>
```

If you want to access enum members by *name*, use item access:

```
>>> Color['red']
<Color.red: 1>
>>> Color['green']
<Color.green: 2>
```

If have an enum member and need its `name` or `value`:

```
>>> member = Color.red
>>> member.name
'red'
>>> member.value
1
```

# Duplicating enum members and values

Having two enum members (or any other attribute) with the same name is invalid; in Python 3.x this would raise an error, but in Python 2.x the second member simply overwrites the first:

```
>>> # python 2.x
>>> class Shape(Enum):
...     square = 2
...     square = 3
...
>>> Shape.square
<Shape.square: 3>

>>> # python 3.x
>>> class Shape(Enum):
...     square = 2
...     square = 3
Traceback (most recent call last):
...
TypeError: Attempted to reuse key: 'square'
```

However, two enum members are allowed to have the same value. Given two members A and B with the same value (and A defined first), B is an alias to A. By-value lookup of the value of A and B will return A. By-name lookup of B will also return A:

```
>>> class Shape(Enum):
...     __order__ = 'square diamond circle alias_for_square'  # only needed in 2.x
...     square = 2
...     diamond = 1
...     circle = 3
...     alias_for_square = 2
...
>>> Shape.square
```

```
<Shape.square: 2>
>>> Shape.alias_for_square
<Shape.square: 2>
>>> Shape(2)
<Shape.square: 2>
```

Allowing aliases is not always desirable. `unique` can be used to ensure that none exist in a particular enumeration:

```
>>> from enum import unique
>>> @unique
... class Mistake(Enum):
...     __order__ = 'one two three four'  # only needed in 2.x
...     one = 1
...     two = 2
...     three = 3
...     four = 3
Traceback (most recent call last):
...
ValueError: duplicate names found in <enum 'Mistake'>: four -> three
```

Iterating over the members of an enum does not provide the aliases:

```
>>> list(Shape)
[<Shape.square: 2>, <Shape.diamond: 1>, <Shape.circle: 3>]
```

The special attribute `__members__` is a dictionary mapping names to members. It includes all names defined in the enumeration, including the aliases:

```
>>> for name, member in sorted(Shape.__members__.items()):
...     name, member
...
('alias_for_square', <Shape.square: 2>)
('circle', <Shape.circle: 3>)
('diamond', <Shape.diamond: 1>)
('square', <Shape.square: 2>)
```

The `__members__` attribute can be used for detailed programmatic access to the enumeration members. For example, finding all the aliases:

```
>>> [name for name, member in Shape.__members__.items() if member.name != name]
['alias_for_square']
```

# Comparisons

Enumeration members are compared by identity:

```
>>> Color.red is Color.red
True
>>> Color.red is Color.blue
False
>>> Color.red is not Color.blue
True
```

Ordered comparisons between enumeration values are *not* supported. Enum members are not integers (but see IntEnum below):

```
>>> Color.red < Color.blue
Traceback (most recent call last):
  File "<stdin>", line 1, in <module>
TypeError: unorderable types: Color() < Color()
```

> ### *Warning*
>
> In Python 2 *everything* is ordered, even though the ordering may not make sense. If you want your enumerations to have a sensible ordering check out the OrderedEnum recipe below.

Equality comparisons are defined though:

```
>>> Color.blue == Color.red
False
>>> Color.blue != Color.red
True
>>> Color.blue == Color.blue
True
```

Comparisons against non-enumeration values will always compare not equal (again, `IntEnum` was explicitly designed to behave differently, see below):

```
>>> Color.blue == 2
False
```

# Allowed members and attributes of enumerations

The examples above use integers for enumeration values. Using integers is short and handy (and provided by default by the Functional API), but not strictly enforced. In the vast majority of use-cases, one doesn't care what the actual value of an enumeration is. But if the value *is* important, enumerations can have arbitrary values.

Enumerations are Python classes, and can have methods and special methods as usual. If we have this enumeration:

```
>>> class Mood(Enum):
...     funky = 1
...     happy = 3
...
...     def describe(self):
...         # self is the member here
...         return self.name, self.value
...
...     def __str__(self):
...         return 'my custom str! {0}'.format(self.value)
...
...     @classmethod
...     def favorite_mood(cls):
```

```
...        # cls here is the enumeration
...        return cls.happy
```

Then:

```
>>> Mood.favorite_mood()
<Mood.happy: 3>
>>> Mood.happy.describe()
('happy', 3)
>>> str(Mood.funky)
'my custom str! 1'
```

The rules for what is allowed are as follows: _sunder_ names (starting and ending with a single underscore) are reserved by enum and cannot be used; all other attributes defined within an enumeration will become members of this enumeration, with the exception of *__dunder__* names and descriptors (methods are also descriptors).

Note:

If your enumeration defines __new__ and/or __init__ then whatever value(s) were given to the enum member will be passed into those methods. See Planet for an example.

# Restricted subclassing of enumerations

Subclassing an enumeration is allowed only if the enumeration does not define any members. So this is forbidden:

```
>>> class MoreColor(Color):
...     pink = 17
Traceback (most recent call last):
...
TypeError: Cannot extend enumerations
```

But this is allowed:

```
>>> class Foo(Enum):
...     def some_behavior(self):
...         pass
...
>>> class Bar(Foo):
...     happy = 1
...     sad = 2
...
```

Allowing subclassing of enums that define members would lead to a violation of some important invariants of types and instances. On the other hand, it makes sense to allow sharing some common behavior between a group of enumerations. (See OrderedEnum for an example.)

# Pickling

Enumerations can be pickled and unpickled:

```
>>> from enum.test_enum import Fruit
>>> from pickle import dumps, loads
```

```
>>> Fruit.tomato is loads(dumps(Fruit.tomato, 2))
True
```

The usual restrictions for pickling apply: picklable enums must be defined in the top level of a module, since unpickling requires them to be importable from that module.

Note:

With pickle protocol version 4 (introduced in Python 3.4) it is possible to easily pickle enums nested in other classes.

# Functional API

The `Enum` class is callable, providing the following functional API:

```
>>> Animal = Enum('Animal', 'ant bee cat dog')
>>> Animal
<enum 'Animal'>
>>> Animal.ant
<Animal.ant: 1>
>>> Animal.ant.value
1
>>> list(Animal)
[<Animal.ant: 1>, <Animal.bee: 2>, <Animal.cat: 3>, <Animal.dog: 4>]
```

The semantics of this API resemble `namedtuple`. The first argument of the call to `Enum` is the name of the enumeration.

The second argument is the *source* of enumeration member names. It can be a whitespace-separated string of names, a sequence of names, a sequence of 2-tuples with key/value pairs, or a mapping (e.g. dictionary) of names to values. The last two options enable assigning arbitrary values to enumerations; the others auto-assign increasing integers starting with 1. A new class derived from `Enum` is returned. In other words, the above assignment to `Animal` is equivalent to:

```
>>> class Animals(Enum):
...     ant = 1
...     bee = 2
...     cat = 3
...     dog = 4
```

Pickling enums created with the functional API can be tricky as frame stack implementation details are used to try and figure out which module the enumeration is being created in (e.g. it will fail if you use a utility function in separate module, and also may not work on IronPython or Jython). The solution is to specify the module name explicitly as follows:

```
>>> Animals = Enum('Animals', 'ant bee cat dog', module=__name__)
```

# Derived Enumerations

## IntEnum

A variation of `Enum` is provided which is also a subclass of `int`. Members of an `IntEnum` can be compared to integers; by extension, integer enumerations of different types can also be compared to each other:

```
>>> from enum import IntEnum
>>> class Shape(IntEnum):
...     circle = 1
...     square = 2
...
>>> class Request(IntEnum):
...     post = 1
...     get = 2
...
>>> Shape == 1
False
>>> Shape.circle == 1
True
>>> Shape.circle == Request.post
True
```

However, they still can't be compared to standard `Enum` enumerations:

```
>>> class Shape(IntEnum):
...     circle = 1
...     square = 2
...
>>> class Color(Enum):
...     red = 1
...     green = 2
...
>>> Shape.circle == Color.red
False
```

`IntEnum` values behave like integers in other ways you'd expect:

```
>>> int(Shape.circle)
1
>>> ['a', 'b', 'c'][Shape.circle]
'b'
>>> [i for i in range(Shape.square)]
[0, 1]
```

For the vast majority of code, `Enum` is strongly recommended, since `IntEnum` breaks some semantic promises of an enumeration (by being comparable to integers, and thus by transitivity to other unrelated enumerations). It should be used only in special cases where there's no other choice; for example, when integer constants are replaced with enumerations and backwards compatibility is required with code that still expects integers.

## Others

While `IntEnum` is part of the `enum` module, it would be very simple to implement independently:

```
class IntEnum(int, Enum):
    pass
```

This demonstrates how similar derived enumerations can be defined; for example a `StrEnum` that mixes in `str` instead of `int`.

Some rules:

1. When subclassing `Enum`, mix-in types must appear before `Enum` itself in the sequence of bases, as in the `IntEnum` example above.

2. While `Enum` can have members of any type, once you mix in an additional type, all the members must have values of that type, e.g. `int` above. This restriction does not apply to mix-ins which only add methods and don't specify another data type such as `int` or `str`.

3. When another data type is mixed in, the `value` attribute is *not the same* as the enum member itself, although it is equivalant and will compare equal.

4. %-style formatting: `%s` and `%r` call `Enum`'s `__str__` and `__repr__` respectively; other codes (such as `%i` or `%h` for IntEnum) treat the enum member as its mixed-in type.

   Note: Prior to Python 3.4 there is a bug in `str`'s %-formatting: `int` subclasses are printed as strings and not numbers when the `%d`, `%i`, or `%u` codes are used.

5. `str.__format__` (or `format`) will use the mixed-in type's `__format__`. If the `Enum`'s `str` or `repr` is desired use the `!s` or `!r` `str` format codes.

# Decorators

## unique

A `class` decorator specifically for enumerations. It searches an enumeration's `__members__` gathering any aliases it finds; if any are found `ValueError` is raised with the details:

```
>>> @unique
... class NoDupes(Enum):
...     first = 'one'
...     second = 'two'
...     third = 'two'
Traceback (most recent call last):
...
ValueError: duplicate names found in <enum 'NoDupes'>: third -> second
```

# Interesting examples

While `Enum` and `IntEnum` are expected to cover the majority of use-cases, they cannot cover them all. Here are recipes for some different types of enumerations that can be used directly, or as examples for creating one's own.

## AutoNumber

Avoids having to specify the value for each enumeration member:

```
>>> class AutoNumber(Enum):
...     def __new__(cls):
...         value = len(cls.__members__) + 1
...         obj = object.__new__(cls)
...         obj._value_ = value
...         return obj
...
>>> class Color(AutoNumber):
...     __order__ = "red green blue"  # only needed in 2.x
...     red = ()
...     green = ()
...     blue = ()
```

```
...
>>> Color.green.value == 2
True
```

Note:

> The _new_ method, if defined, is used during creation of the Enum members; it is then replaced by Enum's _new_ which is used after class creation for lookup of existing members. Due to the way Enums are supposed to behave, there is no way to customize Enum's _new_.

## UniqueEnum

Raises an error if a duplicate member name is found instead of creating an alias:

```
>>> class UniqueEnum(Enum):
...     def __init__(self, *args):
...         cls = self.__class__
...         if any(self.value == e.value for e in cls):
...             a = self.name
...             e = cls(self.value).name
...             raise ValueError(
...                     "aliases not allowed in UniqueEnum:  %r --> %r"
...                     % (a, e))
...
>>> class Color(UniqueEnum):
...     red = 1
...     green = 2
...     blue = 3
...     grene = 2
Traceback (most recent call last):
...
ValueError: aliases not allowed in UniqueEnum:  'grene' --> 'green'
```

## OrderedEnum

An ordered enumeration that is not based on `IntEnum` and so maintains the normal `Enum` invariants (such as not being comparable to other enumerations):

```
>>> class OrderedEnum(Enum):
...     def __ge__(self, other):
...         if self.__class__ is other.__class__:
...             return self._value_ >= other._value_
...         return NotImplemented
...     def __gt__(self, other):
...         if self.__class__ is other.__class__:
...             return self._value_ > other._value_
...         return NotImplemented
...     def __le__(self, other):
...         if self.__class__ is other.__class__:
...             return self._value_ <= other._value_
...         return NotImplemented
...     def __lt__(self, other):
...         if self.__class__ is other.__class__:
...             return self._value_ < other._value_
...         return NotImplemented
```

```
...
>>> class Grade(OrderedEnum):
...     __ordered__ = 'A B C D F'
...     A = 5
...     B = 4
...     C = 3
...     D = 2
...     F = 1
...
>>> Grade.C < Grade.A
True
```

## Planet

If `__new__` or `__init__` is defined the value of the enum member will be passed to those methods:

```
>>> class Planet(Enum):
...     MERCURY = (3.303e+23, 2.4397e6)
...     VENUS   = (4.869e+24, 6.0518e6)
...     EARTH   = (5.976e+24, 6.37814e6)
...     MARS    = (6.421e+23, 3.3972e6)
...     JUPITER = (1.9e+27,   7.1492e7)
...     SATURN  = (5.688e+26, 6.0268e7)
...     URANUS  = (8.686e+25, 2.5559e7)
...     NEPTUNE = (1.024e+26, 2.4746e7)
...     def __init__(self, mass, radius):
...         self.mass = mass        # in kilograms
...         self.radius = radius    # in meters
...     @property
...     def surface_gravity(self):
...         # universal gravitational constant  (m3 kg-1 s-2)
...         G = 6.67300E-11
...         return G * self.mass / (self.radius * self.radius)
...
>>> Planet.EARTH.value
(5.976e+24, 6378140.0)
>>> Planet.EARTH.surface_gravity
9.802652743337129
```

# How are Enums different?

Enums have a custom metaclass that affects many aspects of both derived Enum classes and their instances (members).

## Enum Classes

The `EnumMeta` metaclass is responsible for providing the `__contains__`, `__dir__`, `__iter__` and other methods that allow one to do things with an `Enum` class that fail on a typical class, such as `list(Color)` or `some_var in Color`. `EnumMeta` is responsible for ensuring that various other methods on the final `Enum` class are correct (such as `__new__`, `__getnewargs__`, `__str__` and `__repr__`).

> **Note**
>
> `__dir__` is not changed in the Python 2 line as it messes up some of the decorators included in the stdlib.

## Enum Members (aka instances)

The most interesting thing about Enum members is that they are singletons. `EnumMeta` creates them all while it is creating the `Enum` class itself, and then puts a custom `__new__` in place to ensure that no new ones are ever instantiated by returning only the existing member instances.

## Finer Points

`Enum` members are instances of an `Enum` class, and even though they are accessible as *EnumClass.member1.member2*, they should not be accessed directly from the member as that lookup may fail or, worse, return something besides the `Enum` member you were looking for (changed in version 1.1.1):

```
>>> class FieldTypes(Enum):
...     name = 1
...     value = 2
...     size = 3
...
>>> FieldTypes.value.size
<FieldTypes.size: 3>
>>> FieldTypes.size.value
3
```

The `__members__` attribute is only available on the class.

In Python 3.x `__members__` is always an `OrderedDict`, with the order being the definition order. In Python 2.7 `__members__` is an `OrderedDict` if `__order__` was specified, and a plain `dict` otherwise. In all other Python 2.x versions `__members__` is a plain `dict` even if `__order__` was specified as the `OrderedDict` type didn't exist yet.

If you give your `Enum` subclass extra methods, like the Planet class above, those methods will show up in a *dir* of the member, but not of the class:

```
>>> dir(Planet)
['EARTH', 'JUPITER', 'MARS', 'MERCURY', 'NEPTUNE', 'SATURN', 'URANUS',
'VENUS', '__class__', '__doc__', '__members__', '__module__']
>>> dir(Planet.EARTH)
['__class__', '__doc__', '__module__', 'name', 'surface_gravity', 'value']
```

A `__new__` method will only be used for the creation of the `Enum` members -- after that it is replaced. This means if you wish to change how `Enum` members are looked up you either have to write a helper function or a `classmethod`.