[Reviews · NeurIPS 2019]

Reviewer 1



The approach works for cases where a given agent-task pairing defines a fixed policy, so it's hierarchical in nature. It formulates planning as an agent-task constrained matching problem with singleton and pairwise terms, optimized by a constrained LP or QP. The formulation is scalable (polynomial in nature) and generalizes to larger variants of the scene. The novelty: -- The pairwise agent-task formulation is similar to [22], but focused on agent-task matching in the policy network. 2 different score schemes (Direct and PEM), the latter of which somewhat novel. -- The idea that through structured matching, training from smaller tasks, generalizes to larger tasks and evidence to this effect. -- Experimental results demonstrating intuitive results, that for matching problems QUAD+simpler DM or LP+simpler DM features are superior to PEM features. PEM has higher expressivity, but does not learn as well and does not generalize as well. Rather compelling videos of the resulting strategies are shared in complementary material. The paper is clear, all details are described and code is shared. It would help a bit to elaborate on why exactly AMAX-Pem and Quad-PEM did not converge. Is that a result of the optimization or the model collapsing, or both?

Reviewer 2



Originality: I think in the context of MAC problems the idea of learning a scoring model, and then using it as input to a quadratic or linear program to perform task assingment, may be origional. Quality: Results are adequately presented and analyzed. Clarity: The paper is fairly well written. I think enough information is provided in the paper and supplementary material to allow for reproducibility of results. Significance: The work presented is limited in scope to muti-agent collaboration (MAC) types of problems, but has some useful real world applications.

Reviewer 3



This paper proposes a new multi-task hierarchical reinforcement learning algorithm. The high-level policy achieves the assignment of tasks by solving a linear programming problem(or a quadratic programming problem), and the low-level policy is pre-defined. The biggest contribution of this paper is to get rid of the limitation of the number of agents and the number of tasks by modeling the multi-task assignment problem as an optimization problem, which based on the correlation between the agent and the task and the correlation between the tasks. After training the correlation in a simple task, you only need to re-solve the optimization problem in the complex task, without retraining, thus achieving zero-shot generalization. Contributions: 1. In this paper, the collaboration patterns between agents in the multi-task problem, such as creating subgroups of agents or spreading agents across tasks at the same time, are transformed into constraints to be added to the optimization problem corresponding to the high-level policy. Thereby achieving better collaboration between agents. 2. In order to learn the relationship between agents and tasks, and between tasks and tasks, this paper proposes a variety of learning models. These learning models are implemented using a reinforcement learning algorithm that models the correlation as a continuous action space in an MDP. 3. Due to the different degrees of inductive bias introduced, the fitting ability and generalization ability of relationship learning models are also different. Through rich experiments, this paper verifies the impact of the combination of these models and different optimization problems on the algorithm's ability to fitting ability and generalization ability. But there are some problems in this article. First of all, this paper focuses on the generalization ability of the reinforcement learning algorithm. The conclusion in the paper shows that in order to have better generalization ability, the model with more inductive bias can not be used. Does this affect the final performance of the algorithm after it has been transfered to a new task? Does the algorithm has the ability to continue training on new tasks? Does the algorithm can achieves the performance of the current state of the art algorithms after continuing training? Secondly, the positional embedding model proposed in this paper uses the same embedding learning method as the [35]. The only difference is the subsequent use of these embeddings. The paper should explicitly mention the relationship with the [35] in this section. Finally, this paper introduces the correlated exploration to enhance the exploration of the agents in the multi-agent multi-task environment, but the effectiveness of this method is not verified in the experiment. Here is a small mistake in the paper. The title of [25] was changed to “Learning when to Communicate at Scale in Multiagent Cooperative and Competitive Tasks”.

[Author Response · NeurIPS 2019]

We thank all the reviewers for their insightful reviews. We are going to fix all the reported typos in the final version.

## Reviewer 1

*"why exactly* AMAX *-PEM and* QUAD *-PEM did not converge"* In the Search and Rescue experiments, both AMAX -PEM and QUAD -PEM actually converged to a policy while training. The failures reported in the experiment table refer to the configurations in which the agents do not complete the problem within **????????** steps and they only occur when these models (AMAX -PEM and QUAD -PEM) are evaluated on a different scenario than the one seen in training.

When this kind of failure happens at evaluation time, the algorithm starts oscillating between alternative two or more possible assignments, thus preventing some tasks to be solved (i.e., some victims are never rescued). We hypothesize that this kind of behavior is due to the fact that the receptive fields of the convolutional layers contains a different number of agents and tasks than during training on average, which can cause over or under saturation of the filters, since in this case we would be significantly out of the training support.

## Reviewer 2

*"directly learn assignment policies for test scenarios"* In the attempt of identifying the best performance achievable in different SC scenarios, we ran a test experiment in one of the hard scenarios (w65v67), with a training budget twice as large as the one allocated to the smaller scenario we originally trained on (w15v17). None of the models we introduce in the paper managed to achieve a win-rate above 5%, which is significantly worse than the performance obtained by generalizing from simple to hard. It is indeed possible that the performance we achieved through generalization could be matched or even improved, but it would require much larger computation budgets. Also note that the quadratic problem gets bigger and bigger, and while this does not prevent to run in real-time at evaluation time it does slow down the training significantly.

## Reviewer 3

*Does the algorithm can achieves the performance of the current state of the art algorithms after continuing training?* We do not claim to obtain the best performance achievable in the bigger scenarios in SC, as a topline is not available. As mentioned to Rev.2, achieving a satisfactory performance in hard scenarios seems extremely hard, and to the best of our knowledge our generalization approach is the most promising direction to solve complex MAC problems, which are currently out of reach of state-of-the-art algorithms.

*Does the algorithm has the ability to continue training on new tasks?* This is an interesting question and a direction worth investigating as future work. In this paper, we focused on enabling zero-shot transfer, which we believe is of practical interest, for example when you can't afford to train in the target environment.

*...the model with more inductive bias can not be used. Does this affect the final performance of the algorithm after it has been transfered to a new task?* Although we did not run enough tests to provide a conclusive answer, we believe that the DM should still be superior to PEM even when fine-tuning after generalization is performed. While PEM may allow expressing richer scoring functions, our results show that coordination procedures such as LP or QP are sophisticated enough to allow representing rich coordination patterns.

*Correlated exploration... effectiveness of this method is not verified in the experiment.* The idea of using auto-correlated noise for exploration in continuous action spaces is not novel. Although its application to MAC may be novel, it has been used in DDPG before [Lillicrap et al.], and thus we did not put too much focus on it. As a concrete example on the necessity of this, consider the search and rescue task: the reward is completely non-informative (-1 at each step) and does not guide the agent towards solving all the tasks. This creates an exploration problem since uncorrelated exploration is not likely to solve all the tasks even once because the agents will keep oscillating between tasks. As a result, in our experiments, we have never been able to learn any meaningful policy without it.

## References

Timothy P. Lillicrap, Jonathan J. Hunt, Alexander Pritzel, Nicolas Heess, Tom Erez, Yuval Tassa, David Silver, and Daan Wierstra. Continuous control with deep reinforcement learning.


[Meta-Review · NeurIPS 2019]

The reviewers unanimously recommend accept.